# Effect of Differing Durations of High-Intensity Intermittent Activity on Cognitive Function in Adolescents

**DOI:** 10.3390/ijerph182111594

**Published:** 2021-11-04

**Authors:** Lorna M. Hatch, Karah J. Dring, Ryan A. Williams, Caroline Sunderland, Mary E. Nevill, Simon B. Cooper

**Affiliations:** Department of Sports Science, School of Science and Technology, Nottingham Trent University, Nottingham NG11 8NS, UK; lorna.hatch2014@my.ntu.ac.uk (L.M.H.); karah.dring@ntu.ac.uk (K.J.D.); ryan.williams@ntu.ac.uk (R.A.W.); caroline.sunderland@ntu.ac.uk (C.S.); mary.nevill@ntu.ac.uk (M.E.N.)

**Keywords:** exercise, physical activity, high-intensity, duration, attention, inhibitory control, working memory, cognition, adolescents

## Abstract

Exercise duration may influence the acute effects on cognition. However, only one study to date has explored the dose-response relationship between exercise duration and cognition in adolescents. Thus, the purpose of this study was to investigate the effect of differing durations of high-intensity intermittent running on cognition in adolescents. Thirty-eight adolescents (23 girls) completed three trials separated by 7 d: 30 min exercise, 60 min exercise, and rest; in a randomised crossover design. The exercise was a modified version of the Loughborough Intermittent Shuttle Test (LIST), which elicited high-intensity intermittent exercise. Cognitive function tests (Stroop test, Sternberg paradigm, Flanker task) were completed 30 min pre, immediately post, and 45 min post exercise. Response times on the incongruent level of the Flanker task improved to a greater extent 45 min following the 30 min LIST, compared to rest (*p* = 0.009). Moreover, response times improved to a greater extent on the three-item level of Sternberg paradigm 45 min following the 30 min LIST, compared to the 60 min LIST (*p* = 0.002) and rest (*p* = 0.013), as well as on the five-item level 45 min following the 30 min LIST, compared to the 60 min LIST (*p* = 0.002). In conclusion, acute exercise enhanced subsequent cognition in adolescents, but overall, 30 min of high-intensity intermittent running is more favourable to adolescents’ cognition, compared to 60 min.

## 1. Introduction

The existing evidence suggests that an acute bout of exercise can improve subsequent cognitive function across a range of domains in young people, including attentional capacity [1], executive function [2], and working memory [3]. These cognitive processes are responsible for self-regulation and goal-orientated behaviours [4] and are fundamental to learning [5]. Therefore, participation in physical activity has the potential to improve cognitive performance and academic achievement in young people [6]. Moreover, current literature highlights several factors that may mediate the exercise-cognition relationship, including an increase in cerebral blood flow [7], neurogenesis [8], and activation of brain regions involved in cognitive processes (e.g., cerebellum, prefrontal cortex) [1,9,10].

The majority of the literature on the acute effects of exercise on cognition in adolescents has employed running and cycling modalities that were continuous in nature (for review, see the work of [11]). Research demonstrates, however, that young people’s activity patterns are typically intermittent in nature, involving short bursts of high-intensity activity interspersed with rest [12,13], and rarely consist of sustained moderate or vigorous activity [14]. Additionally, high-intensity intermittent exercise is enjoyable to youth [15], which is a particularly important consideration when looking to develop ecologically valid forms of physical activity with the aim of achieving long-term, sustained behaviour change [13]. One of the few exercise-cognition studies to use high-intensity intermittent exercise in adolescents found that both working memory and executive function were improved following 60 min of games-based basketball activity, compared to rest, and that enhancements to executive function lasted 45 min following the exercise [16]. Additionally, high-intensity intermittent sprinting (10 × 10 s running sprints) has been shown to enhance adolescents’ inhibitory control and information processing [17]. High-intensity intermittent exercise may thus provide an ecologically valid, efficacious type of exercise for enhancing cognition in youth.

The exercise-cognition relationship is, however, a complicated one, and several review papers and meta-analyses highlight that the characteristics of exercise, such as the modality, intensity, and duration, have a moderating effect on the subsequent effects on cognition [18,19,20,21,22]. Moreover, the characteristics of an exercise are inherently linked, meaning the duration of an exercise will interact with the modality and intensity to determine the overall dose [22]. While a wide range of exercise modalities, intensities, and durations have been used across the literature, very few studies have systematically compared exercise of different characteristics (e.g., modality, intensity, and duration) within the same study. Such studies would provide invaluable insight into how to optimise the cognitive benefits following exercise in young people.

One such key variable in the exercise-cognition relationship is exercise duration. In particular, establishing the minimum duration of exercise required for improvements to cognition may be particularly useful for school staff and policymakers, who are keen to support young people’s learning through exercise, but frequently cite time constraints as a barrier hindering the implementation of exercise in schools [23,24,25]. A systematic review on the effects of acute exercise on attention in youth (aged 4–18 years old) noted that short exercise bouts (≤20 min) had a positive effect on attention, while some longer exercise bouts (e.g., 45 min) had no effect [20]. A meta-analytic review on adults and youth, however, concluded that exercise of short durations (≤10 min) had a negligible effect on cognitive performance, while exercise lasting > 11 min had positive effects [18]. Moreover, a recent review of the child and adolescent literature concluded that exercise ~30 min duration has positive effects across cognitive domains in children and that exercise lasting 10–30 min is most beneficial in adolescents [22]. The heterogeneity in the conclusions of reviews and meta-analyses results from the heterogeneity between the studies, which have been used to make these conclusions. The studies adopt different modalities and intensities of exercise, and different cognitive outcome measures, all of which make it difficult to directly compare the effects of exercise bouts of different durations with each other. A more informed understanding of the exercise duration-cognition relationship can only begin to be established once there is sufficient primary research that directly compares exercise of differing durations within the same study (while holding other key variables, such as intensity and modality, constant).

Of the few studies to use multiple durations of exercise, two were conducted by Howie and colleagues [26,27]. Both studies examined the effect of 5, 10, and 20 min of moderate-to-vigorous classroom-based exercise, compared to 10 min of sedentary activity, on the cognitive performance of children (aged 9–12 years) [26,27]. Children presented higher math fluency scores after 10 and 20 min of exercise when compared to 10 min of sedentary activity [27]. Moreover, on-task behaviour was improved after 10 min of exercise, and there was a trend towards improved on-task behaviour after 20 min of exercise, again when compared to the sedentary condition [26]. In contrast, no improvements in executive function or working memory were evident after any of the exercise bouts [27]. However, while these studies investigated the effect of different exercise durations, separate analyses were conducted, meaning the effects of each duration of exercise were not compared to one another, thus limiting the conclusions that can be drawn.

To date, only one study has directly compared the effects of exercise of differing durations on cognitive function in adolescents. The study compared the effects of 10, 20, and 30 min of moderate-intensity (40–60% of heart rate reserve) cycling on adolescents’ (11–14 years) selective attention and working memory [28]. The authors reported no effect on selective attention or working memory performance following any duration of exercise compared to a time-matched resting control. Moreover, no differential effects of the exercise durations on post-exercise cognition was observed. However, a between-subjects design was used to compare the exercise durations, with a different group of children performing each duration of exercise. Individual differences between the groups, such as in baseline cognitive ability and cardiorespiratory fitness, may have thus influenced the results [22]. Additionally, it is possible that the cycling intensity used in the study was not high enough to elicit a cognitive response when completed for ≤30 min. This is also in accordance with the wider adolescent literature, where, for example, no effects to executive function following cycling at a light intensity (60% heart rate max) for 20 min were observed, but enhancements to executive function following incremental cycling to exhaustion were reported [29].

Therefore, the present study aims to examine the effects of 30 min and 60 min of high-intensity intermittent running, compared to rest, on immediate and delayed (45 min post exercise) cognitive function in adolescents. The study thus builds on previous research by using an ecologically valid modality of exercise and by directly comparing multiple durations of exercise using a within-subjects, randomised crossover design. Based on the literature to date, the hypothesis of the present study is that high-intensity intermittent running will enhance subsequent cognition, while the comparison of 30 and 60 min high-intensity intermittent running is exploratory.

## 2. Materials and Methods

### 2.1. Participant Characteristics

Forty-one participants were recruited to participate in the study. However, based on the exclusion criteria, three participants were removed from the study due to the presence of a congenital heart condition (*n* = 1) and an inability to complete the 60 min of exercise (*n* = 2). Therefore, thirty-eight participants (15 boys, 23 girls) completed the study.

During familiarisation, body mass (Seca 770 digital scale, Hamburg, Germany), stature, and sitting stature (Leicester Height Measure, Seca, Hamburg, Germany) were measured. These measures were subsequently used to calculate body mass index (BMI) and age at peak height velocity [30]. Moreover, waist circumference was measured, and four skinfold measurements (tricep, subscapular, supraspinale, and front thigh) were taken, according to the International Society for the Advancement of Kinanthropometry procedures. For descriptive purposes, anthropometric characteristics are displayed in Table 1.

### 2.2. Study Design

Following approval from the University ethical advisory committee, participants were recruited from local secondary schools in the East Midlands, U.K. Consent from the headteacher of each school was acquired. Following this, participants were recruited from the school on a voluntary basis. Participants provided their written assent, and parents/guardians provided their written consent. Parents/guardians also completed a health screen questionnaire on behalf of the participant to ensure that there were no health conditions that would affect their ability to complete the study (e.g., heart conditions, uncontrolled exercise-induced asthma).

The study employed a within-subject, randomised, counterbalanced, crossover design that involved children completing a familiarisation session followed by three main trials (30 min high-intensity intermittent running, 60 min high-intensity intermittent running, and rest), which were each separated by seven days. The Latin Square Design was used to randomise the order in which participants completed the three main trials.

During familiarisation, the experimental protocols were explained to participants, and participants practiced the procedures to be completed during the main trials. This included the battery of cognitive function tests and high-intensity intermittent exercise. The exercise protocol employed for this study was the Loughborough Intermittent Shuttle Test (LIST). This exercise protocol enables other exercise characteristics (e.g., intensity) and external factors (e.g., physical environment, cognitive engagement) to be controlled while manipulating the exercise duration to explore the effects on cognition. This level of control is not possible with other types of intermittent exercise such as games-based exercise. Furthermore, during familiarisation, participants also completed the Multi-Stage Fitness Test (MSFT) [31].

Prior to the first main trial, participants recorded a food diary for 24 h. Participants were asked to repeat their recorded diets prior to each subsequent experimental trial. Participants fasted from 9 p.m. the evening prior to each experimental trial and refrained from physical activity 24 h prior to and during all experimental trials (with the exception of during the LIST protocol). Water was allowed ad libitum.

On arrival to school (~8.30 am) on the day of each experimental trial, participants were fitted with a heart rate monitor (Team Sports System, Firstbeat Technologies Ltd., Jyvaskyla, Finland), which was worn throughout the trial. As both breakfast consumption [32] and composition [33] affect young people’s subsequent cognitive functioning, participants were provided with a standardised breakfast (cornflakes, milk, and toast with margarine) that contained 1.5 g carbohydrate per kg body mass, as used successfully in studies with adolescents [16,34,35]. Participants had 15 min to consume the standardised breakfast. Figure 1 displays a schematic of the experimental protocol.

#### 2.2.1. Multi-Stage Fitness Test (MSFT)

During familiarisation, participants completed the MSFT, which assesses cardiorespiratory fitness using an incremental exercise protocol whereby participants run at a gradually increasing intensity until volitional exhaustion [31]. The MSFT is a valid and reliable measurement of cardiorespiratory fitness [36,37], which has been used successfully in similar field-based studies in youth [38,39]. In the present study, shuttle level attained in the MSFT was used to estimate peak oxygen consumption (VO_2_ peak), using an adolescent-specific equation [40]. VO_2_ peak was subsequently used to determine the running speeds for the LIST exercise protocol.

#### 2.2.2. Exercise Protocol

During the exercise trials, participants completed either 30 min or 60 min of high-intensity intermittent running in an adapted form of the Loughborough Intermittent Shuttle Test (LIST) [41]. The LIST was conducted in each participating school’s sports hall and involved participants running between two markers, 20 m apart, to pre-determined speeds dictated by an audio signal. The LIST protocol in the present study consisted of three 20 m shuttles at walking pace, a 15 m sprint followed by rest (8 s total duration), three 20 m shuttles running at 85% VO_2_ peak, and three 20 m shuttles running at 55% of VO_2_ peak. This pattern was repeated eight times, lasting ~12 min; this equaled one block (as presented in Figure 2). The 30 min trial consisted of 2 of these blocks and the 60 min trial 4 blocks, with a 3 min recovery provided between each block.

#### 2.2.3. Cognitive Function Tests

The battery of cognitive function tests consisted of the Stroop test, Sternberg paradigm, and Flanker task, completed in that order. The Stroop test is a measure of information processing and inhibitory control [42,43], the Sternberg paradigm assesses visual working memory [44], and the Flanker task measures attention and inhibitory control [45]. The test battery was completed 30 min pre, immediately post, and 45 min post exercise, and at the corresponding time points on the resting trial. The tests lasted ~15 min and were administered via a laptop computer (Leno ThinkPad T450; Lenovo, Hong Kong). Instructions were presented on screen and were repeated verbally by an investigator prior to the completion of each test. Questions were encouraged, and then confirmation of understanding was sought from participants before proceeding. Each test (and test level) was preceded by 3–6 practice stimuli, and participants received feedback regarding whether their responses were correct or not. The practice stimuli re-familiarised participants with the test and acted to negate any potential learning effects. Participants completed the tests in a classroom and were seated separately to ensure no interaction during the tests occurred. Sound-cancelling headphones were worn, and the lights were dimmed to minimise external disturbances and enhance screen visibility. Participants were instructed to respond to each test as quickly and as accurately as possible, with the outcome variables of interest for all tests being the response time of correct responses (ms) and the proportion (%) of correct responses made. These cognitive tests and testing protocols have previously been used successfully in a similar study population [16,17,35].

### 2.3. Statistical Analysis

Cognitive function data were analysed using the open-source software, R (www.r-project.org; accessed on 11 February 2021). Prior to analyses, minimum (<200 ms) and maximum (1500–3000 ms depending on task complexity) response time cut-offs were applied to eliminate any unreasonably fast (anticipatory) or slow (distracted) responses, and response time data were log transformed to exhibit the right-hand skew of typical human response times. This method is widely used in similar studies [16,17,35,46]. Response time analyses were then conducted using mixed-effect models, implemented via the *nlme* package, which yields *t* statistics. Accuracy analyses were also analysed using mixed-effect models but using the *lme4* package due to the binomial nature of the accuracy data. This approach yields *z* statistics. All analyses were conducted using a two-way trial by time interaction. Data for each level of the cognitive tests were analysed separately, given that the different levels require different cognitive processes. For all analyses, statistical significance was accepted as *p* ≤0.05. Cognitive data are presented as mean ± standard error of the mean (SEM); all other data are presented as mean ± standard deviation (SD).

## 3. Results

### 3.1. Cognitive Function

The data for each of the cognitive function tests, on each trial and at each time point, can be found in Table 2. For clarity and ease of interpretation, cognitive function data in figures are presented as change across the morning.

#### 3.1.1. Stroop Test

Response Times: On the simple level of the Stroop test, response times improved 45 min post exercise on the 30 min LIST trial, compared to both the resting control trial (trial × time interaction, *t*_(6544)_ = 2.95, *p* = 0.003; Figure 3) and the 60 min LIST trial (trial × time interaction, *t*_(6544)_ = 3.49, *p* = 0.001; Figure 3). There was also a tendency for response times to improve to a greater extent immediately post exercise on the 30 min LIST trial, compared to the resting control trial, but this did not reach statistical significance (trial × time interaction, *t*_(6544)_ = 1.90, *p* = 0.058; Figure 3). The pattern of change in response times across the morning on the complex level of the Stroop test was not different between the trials (trial × time interactions, *p* = 0.212–0.946). Accuracy: Accuracy, on both the simple and complex levels of the Stroop test, was similar across the morning on all trials (trial × time interactions, *p* = 0.123–0.969).

#### 3.1.2. Sternberg Paradigm

*Response Times:* On the one-item level of the Sternberg paradigm, response times slowed immediately following the 30 min LIST compared to both the resting control trial (trial × time interaction, *t*_(5331)_ = −3.14, *p* = 0.002; Figure 4) and the 60 min LIST trial (trial × time interaction, *t*_(5331)_ = −4.36, *p* < 0.001; Figure 4). However, 45 min post exercise, response times were improved following the 30 min LIST compared to the resting control trial (trial × time interaction, *t*_(5331)_ = 3.62, *p* < 0.001; Figure 4), and tended to be improved to a greater extent following 30 min LIST compared to 60 min LIST (trial × time interaction, *t*_(5331)_ = 1.74, *p* = 0.084; Figure 4), and following 60 min LIST compared to the resting control trial (trial × time interaction, *t*_(3522)_ = 1.82, *p* = 0.069; Figure 4).

On the three-item level of the Sternberg paradigm, response times were improved to a greater extent 45 min post exercise on the 30 min LIST trial, compared to both the resting control trial (trial × time interaction, *t*_(10670)_ = 2.49, *p* = 0.013; Figure 5) and the 60 min LIST trial (trial × time interaction, *t*_(10670)_ = 30.7, *p* = 0.002; Figure 5).

On the five-item level of the Sternberg paradigm, response times improved to a greater extent immediately (trial × time interaction, *t*_(6718)_ = −1.96, *p* = 0.050; Figure 6) and 45 min (trial × time interaction, *t*_(10670)_ = −3.22, *p* = 0.001; Figure 6) following resting, compared to the 60 min LIST. Response times were also improved to a greater extent 45 min post exercise on the 30 min LIST trial, compared to the 60 min LIST trial (trial × time interaction, *t*_(10670)_ = −3.07, *p* = 0.002; Figure 6).

*Accuracy:* Accuracy, on all three levels of the Sternberg paradigm, was similar across the morning on all trials (trial × time interactions, *p* = 0.151–0.969), with the exception of the three-item level where accuracy was better maintained 45 min post exercise on the 60 min LIST trial compared to the 30 min LIST trial (trial × time interaction, *z*_(11286)_ = 2.14, *p* = 0.032; Figure 7).

#### 3.1.3. Flanker Task

*Response Times:* Response times on the congruent level of the Flanker task were improved on the 30 min LIST trial immediately (trial × time interaction, *t*_(10070)_ = −2.19, *p* = 0.029; Figure 8) and 45 min (trial * time interaction, *t*_(10070)_ = −3.85, *p <* 0.001; Figure 8) post exercise compared to the resting trial. Furthermore, response times were improved to a greater extent 45 min post exercise on the 30 min LIST, compared to the 60 min LIST, trial (trial × time interaction, *t*_(10070)_ = −2.57, *p* = 0.010; Figure 8).

Response times on the incongruent level of the Flanker task improved to a greater extent 45 min post exercise on the 30 min LIST trial compared to the resting control trial (trial × time interaction, *t*_(9768)_ = −2.61, *p* = 0.009; Figure 9), and there was a tendency for response times to be improved immediately post exercise on the 30 min LIST trial, compared to the 60 min LIST trial (trial × time interaction, *t*_(9768)_ = −1.79, *p* = 0.073; Figure 9).

*Accuracy:* Accuracy, on both levels of the Flanker task, was similar across the morning on all trials (trial × time interactions, *p* = 0.568–0.962), with the exception of the congruent level whereby there was a tendency for accuracy to be improved 45 min post exercise on the 30 min LIST trial compared to the resting control trial (trial × time interaction, *z*_(10404)_ = 1.67, *p* = 0.095; Figure 10).

## 4. Discussion

The present study is the first to directly compare, using a within-subject crossover design, the acute effects of different durations of exercise on cognition in young people. The main finding of the present study was that there were some duration-specific effects on the exercise-cognition relationship, whereby participation in 30 min, compared to 60 min, of high-intensity intermittent exercise was more beneficial to both immediate and delayed (45 min post) cognitive function. This novel finding suggests that high-intensity intermittent exercise may be an effective type of exercise for enhancing cognition in adolescents, even at shorter durations of 30 min. Furthermore, the present study found that cognitive function was enhanced following high-intensity intermittent exercise regardless of duration when compared to following resting.

Firstly, compared to following resting, working memory performance was enhanced following 60 min of high-intensity intermittent exercise; this was evidenced through faster response times immediately post exercise and a tendency for faster response times 45 min post exercise, on the one-item level of the Sternberg paradigm. Moreover, attention and information processing improved immediately and 45 min following the 30 min high-intensity intermittent exercise, compared to rest, as demonstrated by reduced response times on the simple level of the Stroop test and congruent level of the Flanker task, as well as better maintained accuracy on the congruent level of the Flanker task. Furthermore, inhibitory control was also enhanced 45 min following the 30 min high-intensity intermittent exercise, compared to rest, which was evidenced by reduced response times on the incongruent level of the Flanker task. The results of the present study thus demonstrate that participation in an acute bout of high-intensity intermittent exercise is beneficial to both immediate and delayed (45 min post) cognition and is favourable compared to rest regardless of exercise duration.

When comparing the effects of 30 min and 60 min of high-intensity intermittent exercise, differing effects on cognition were observed. Firstly, while accuracy on the three-item level of the Sternberg paradigm decreased following both exercise and rest, it was better maintained 45 min following 60 min, compared to 30 min, of high-intensity intermittent exercise. However, information processing was improved to a greater extent immediately following 30 min, compared to 60 min, of high-intensity intermittent exercise. This was shown through faster response times on the simple level of the Stroop test. Moreover, inhibitory control tended to be better immediately following the 30 min, compared to 60 min, of high-intensity intermittent exercise, as demonstrated by faster response times on the incongruent level of the Flanker task. With regards to 45 min post exercise, 30 min compared to 60 min of high-intensity intermittent exercise led to enhanced information processing, attention, and inhibitory control. This was evidenced not only by faster response times on the simple level of the Stroop test and on the congruent and incongruent level of the Flanker task but also through a tendency for better maintained accuracy on the congruent level of the Flanker task following the 30 min of exercise. Furthermore, working memory performance was also improved 45 min following 30 min, compared to 60 min, of high-intensity intermittent exercise, which was demonstrated through faster response times on all levels of the Sternberg paradigm. These findings suggest that 30 min of high-intensity intermittent exercise may be more effective than 60 min of high-intensity intermittent exercise for enhancing acute cognitive performance in young people.

The key finding that 30 min of high-intensity intermittent exercise was advantageous over 60 min is interesting, given that previous research that compared exercise of multiple durations found no effects to cognition following exercise ≤30 min [26,27,28]. One study, for example, reported no effects on adolescents’ working memory or selective attention from 10, 20, or 30 min of moderate-to-vigorous intensity cycling compared to rest [28]. However, as highlighted in a prominent review, the modality and intensity of the exercise undertaken are also important, as these quantitative characteristics of the exercise interact with the exercise duration to influence the overall ‘dose’ of the activity [11]. Therefore, high-intensity intermittent exercise, such as that employed in the present study, may be a particularly efficacious modality and intensity of exercise for enhancing cognition, even at shorter durations.

Additionally, the finding that 30 min, compared to 60 min, of high-intermittent exercise was more beneficial to adolescents’ cognition may be due to the longer duration being too demanding for some participants. Specifically, while young people typically choose to engage in high-intensity intermittent activity during both discretionary exercise [13,14] and self-paced exercise interventions [38], the majority of young people do not meet the government-recommended 60 min of daily physical activity [47,48]. Therefore, the fewer beneficial effects seen after 60 min high-intensity intermittent exercise may be due to this duration of high-intensity exercise being too physiologically demanding, particularly for less fit adolescents, resulting in high levels of fatigue, which may detrimentally affect cognitive performance [16].

The findings from this study and the other duration studies [26,27,28] suggest that there may be an inverted-U curvilinear relationship between exercise duration and subsequent cognition. Specifically, the available data suggest that the positive effects of exercise on cognition are lesser following shorter (≤20 min) and longer (~60 min) durations of exercise; while greater benefits are seen following medium (~30 min) durations. However, as aforementioned, the modality and intensity of the exercise undertaken are likely to influence the nature of the relationship [11]. Therefore, additional dose-response studies, which use different modalities and intensities of exercise, are necessary to be able to elucidate the full nature of the exercise duration-cognition relationship. Moreover, future research should seek to compare the effects to cognition from participation in shorter durations (e.g., 5, 10, 15, and 20 min) of high-intensity intermittent exercise than used in the present study to establish whether this modality of exercise is superior to others in its ability to enhance cognition, even at shorter durations. Nonetheless, a key novel finding of the present study is that 30 min of high-intensity intermittent exercise was more beneficial to subsequent cognition in adolescents than 60 min of high-intensity intermittent exercise.

The findings of the present study are valuable to schools and policymakers, as attention, inhibitory control, and working memory are fundamental for goal-orientated behaviours, concentration, and learning [5,6,49,50,51], and the findings demonstrate that only 30 min of high-intensity intermittent exercise, which is markedly more feasible to implement in school compared to 60 min of high-intensity intermittent exercise, is required to enhance these cognitions. The findings also reveal that these cognitive processes are enhanced 45 min following the cessation of 30 min high-intensity intermittent exercise. This information can be used to tailor school-based interventions and physical education sessions (e.g., the time at which they are implemented) to support learning and academic performance throughout the school day. Moreover, 30 min high-intensity intermittent exercise, compared to rest, was recently found to lower adolescents’ post-exercise blood glucose concentration and acute post-prandial insulinaemic response, suggesting that young people also gain benefits to cardiometabolic health, such as enhanced insulin sensitivity, from participation in this type and duration of exercise [34]. Therefore, implementing 30 min of high-intensity intermittent exercise into the school day will benefit young people’s cardiometabolic health, as well as their cognition. Furthermore, youth enjoy participating in high-intensity intermittent exercise [15,52]; thus, high levels of investment [53] will assist with long-term adherence and sustained behaviour change [13]. An interesting direction for future research may be investigating the effect of different durations of a cognitively engaging high-intensity intermittent exercise on cognition, as cognitively engaging exercise is a relatively novel subject of research but has been shown to enhance cognition in youth [1] and in the older population [54].

## 5. Conclusions

In conclusion, the present study produced two main novel findings. Firstly, greater benefits to cognition were observed following participation in 30 min, compared to 60 min, of high-intensity intermittent running. This was evidenced by better information processing and inhibitory control immediately and 45 min following 30 min compared to 60 min of exercise, in addition to enhanced attention and working memory 45 min following 30 min compared to 60 min of exercise. Secondly, participation in an acute bout of high-intensity intermittent running enhanced immediate and delayed (45 min post) cognition in adolescents and was advantageous compared to resting, regardless of the exercise duration. Future research should seek to compare the effects of shorter durations (e.g., 5 vs. 10 vs. 20 min) of high-intensity intermittent running on cognition. Moreover, future research should continue to explore the time-course of effects to cognition following high-intensity intermittent running, to establish whether any beneficial effects last beyond 45 min following cessation of the exercise. The findings of the present study and of this future research will be particularly valuable to school staff and policymakers, as high-intensity intermittent exercise could be implemented within the school day to enhance adolescents’ cognition and, subsequently, learning and academic achievement.

## Figures and Tables

**Figure 1 ijerph-18-11594-f001:**
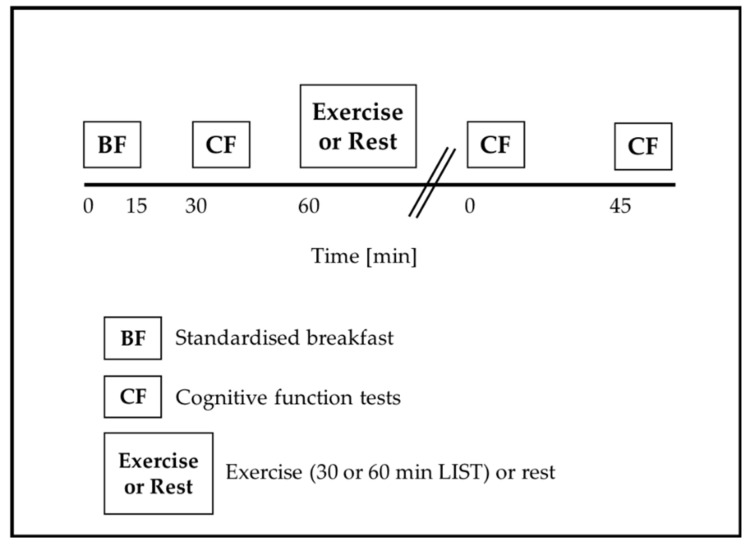
Experimental trial protocol.

**Figure 2 ijerph-18-11594-f002:**
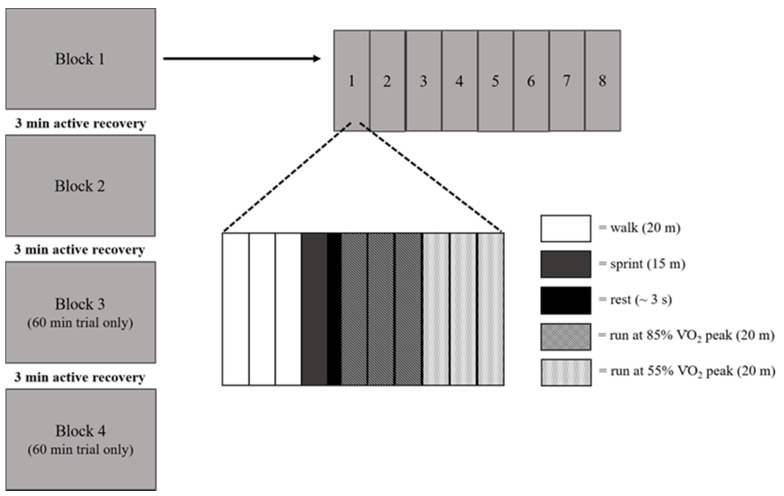
LIST protocol (adapted from Nicholas et al., 2000).

**Figure 3 ijerph-18-11594-f003:**
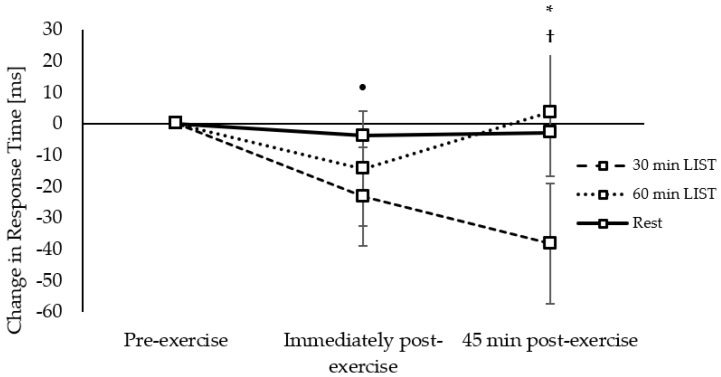
Change in response times across the morning on the 30 min LIST, 60 min LIST, and resting control trials on the simple level of the Stroop test. Faster response times following 30 min LIST vs. 60 min LIST * and rest †; both *p* < 0.05. Tendency for faster response times following 30 min LIST vs. rest; *p* = 0.058 •.

**Figure 4 ijerph-18-11594-f004:**
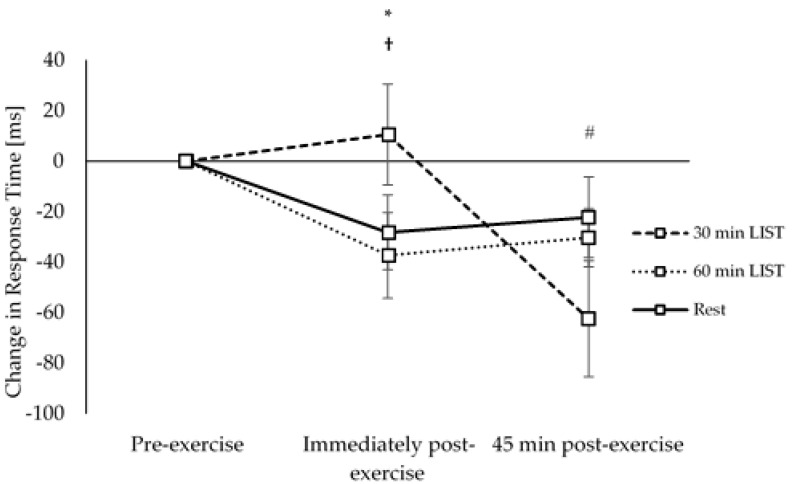
Change in response times across the morning on the 30 min LIST, 60 min LIST, and resting control trials on the one-item level of the Sternberg paradigm. Slower response times following 30 min LIST vs. 60 min LIST * and rest †; faster response times following 30 min LIST vs. rest ^#^; all *p* < 0.05.

**Figure 5 ijerph-18-11594-f005:**
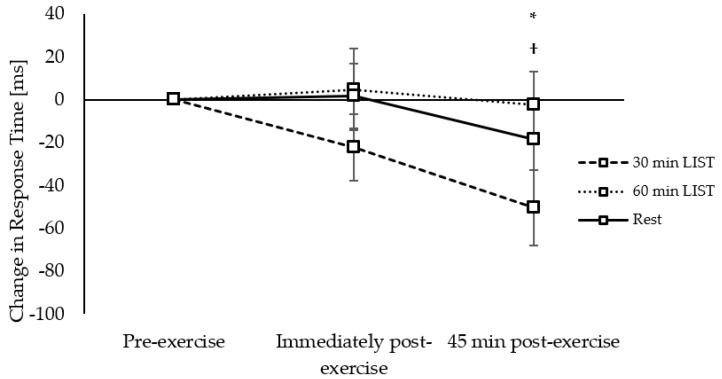
Change in response times across the morning on the 30 min LIST, 60 min LIST, and resting control trials on the three-item level of the Sternberg paradigm. Faster response times following 30 min LIST vs. 60 min LIST * and rest †; all *p* < 0.05.

**Figure 6 ijerph-18-11594-f006:**
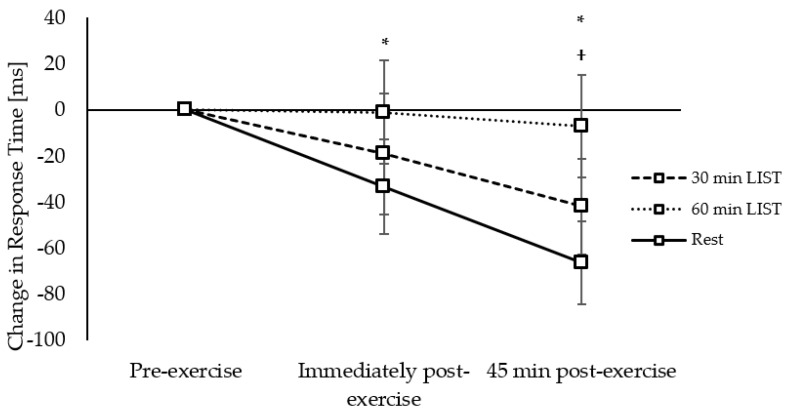
Change in response times across the morning on the 30 min LIST, 60 min LIST, and resting control trials on the five-item level of the Sternberg paradigm. Faster response times following rest vs. 60 min LIST *; faster response times following 30 min LIST vs. 60 min LIST †; all *p* ≤ 0.05.

**Figure 7 ijerph-18-11594-f007:**
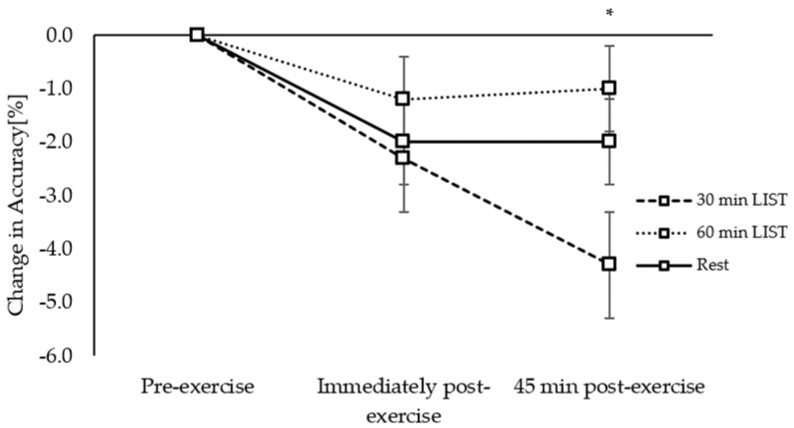
Change in accuracy across the morning on the 30 min LIST, 60 min LIST, and resting control trials on the three-item level of the Sternberg paradigm. Greater accuracy following 60 min vs. 30 min LIST *; *p* < 0.05.

**Figure 8 ijerph-18-11594-f008:**
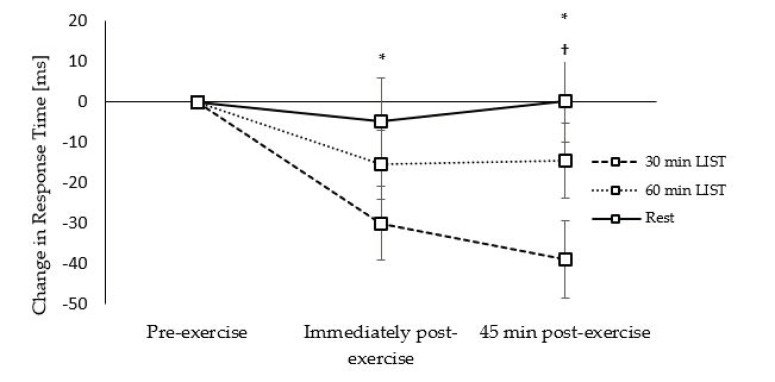
Change in response times across the morning on the 30 min LIST, 60 min LIST, and resting control trials on the congruent level of the Flanker task. Faster response times following 30 min LIST vs. rest * and 60 min LIST †; all *p* < 0.05.

**Figure 9 ijerph-18-11594-f009:**
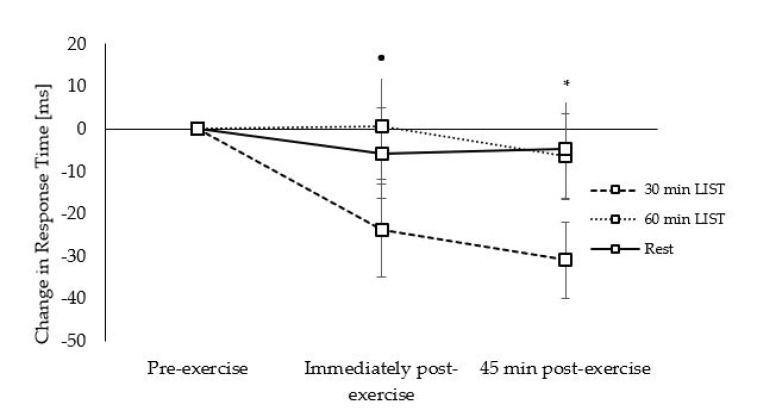
Change in response times across the morning on the 30 min LIST, 60 min LIST, and resting control trials on the incongruent level of the Flanker task. Faster response times following 30 min LIST vs. rest *; *p* < 0.05. Tendency for faster response times following 30 min vs. 60 min LIST •; *p* = 0.073.

**Figure 10 ijerph-18-11594-f010:**
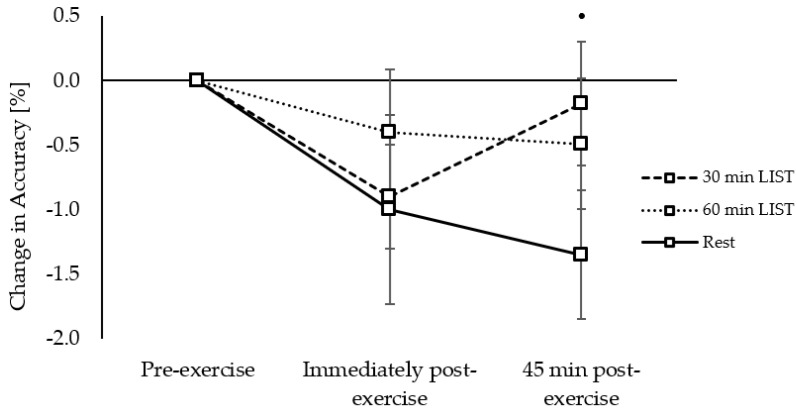
Change in accuracy across the morning on the 30 min LIST, 60 min LIST, and resting control trials on the congruent level of the Flanker task. Tendency for greater accuracy following 30 min LIST vs. rest •; *p* = 0.095.

**Table 1 ijerph-18-11594-t001:** Anthropometric characteristics.

	Overall (*n* = 38)	Boys (*n* = 15)	Girls (*n* = 23)	*p*-Value ^a^
Age (yrs)	12.4 ± 0.4	12.3 ± 0.5	12.4 ± 0.4	0.507
Height (cm)	157.7 ± 7.5	155.9 ± 9.1	159.0 ± 6.2	0.227
Body mass (kg)	45.0 ± 7.2	44.2 ± 7.5	45.5 ± 7.1	0.573
Maturity offset (yrs) ^b^	−1.4 ± 0.6	−1.8 ± 0.4	−1.2 ± 0.6	0.236
Waist circumference (cm)	65.0 ± 4.9	64.5 ± 4.2	65.4 ± 5.4	0.580
Sum of skinfolds (mm)	44.5 ± 13.7	39.9 ± 9.4	47.5 ± 15.4	0.097

Note. ^a^ Independent samples *t*-test for comparison between boys and girls. ^b^ Calculated using the method of [30].

**Table 2 ijerph-18-11594-t002:** Cognitive function data across the morning on the 30 min exercise, 60 min exercise, and resting control trials. Data are mean ± SEM.

Test	Level	Variable	30 min Exercise Trial			60 min Exercise Trial			Resting		
			Pre	Immediately Post	45 min Post	Pre	Immediately Post	45 min Post	Pre	Immediately Post	45 min Post
Stroop test	Simple	Response times (ms)	741 ± 19	717 *±* 24	703 *±* 22	732 *±* 21	718 *±* 26	736 *±* 24	727 *±* 22	723 *±* 24	724 *±* 22
		Accuracy (%)	98.4 *±* 0.5	96.2 *±* 0.8	97.0 *±* 0.8	97.6 *±* 0.6	96.2 *±* 0.8	95.7 *±* 0.9	97.3 *±* 0.5	96.0 *±* 0.7	95.4 *±* 1.3
	Complex	Response times (ms)	1009 *±* 34	974 *±* 37	955 *±* 34	1019 *±* 36	971 *±* 39	972 *±* 44	1003 *±* 37	952 *±* 39	964 *±* 39
		Accuracy (%)	95.4 *±* 0.6	95.7 *±* 0.8	94.5 *±* 1.1	95.9 *±* 0.5	94.6 *±* 0.7	94.4 *±* 0.7	95.1 *±* 0.6	93.4 *±* 1.1	93.7 *±* 1.3
Sternberg paradigm	One-item	Response times (ms)	531 *±* 19	541 *±* 21	468 *±* 15	529 *±* 19	491 *±* 22	499 *±* 22	511 *±* 17	483 *±* 19	489 *±* 16
		Accuracy (%)	97.2 *±* 1.0	97.0 *±* 1.0	94.4 *±* 1.0	97.3 *±* 1.0	97.0 *±* 1.0	96.2 *±* 1.0	96.9 *±* 1.0	95.6 *±* 1.0	95.6 *±* 1.0
	Three-item	Response times (ms)	672 *±* 17	650 *±* 17	621 *±* 16	663 *±* 20	668 *±* 30	660 *±* 36	647 *±* 20	649 *±* 19	629 *±* 21
		Accuracy (%)	97.3 *±* 0.6	95.0 *±* 0.9	93.3 *±* 0.7	96.1 *±* 0.6	94.8 *±* 0.7	95.0 *±* 0.7	96.7 *±* 0.6	94.7 *±* 0.7	94.7 *±* 0.9
	Five-item	Response times (ms)	831 *±* 25	812 *±* 24	789 *±* 17	808 *±* 28	807 *±* 30	801 *±* 31	823 *±* 23	789 *±* 22	757 *±* 22
		Accuracy (%)	94.7 *±* 0.8	91.6 *±* 1.1	91.9 *±* 1.0	93.8 *±* 0.8	92.1 *±* 1.2	92.3 *±* 1.1	92.5 *±* 1.1	91.5 *±* 1.0	89.9 *±* 1.8
Flanker task	Congruent	Response times (ms)	559 *±* 15	529 *±* 14	521 *±* 14	567 *±* 19	552 *±* 21	553 *±* 21	548 *±* 15	544 *±* 18	549 *±* 17
		Accuracy (%)	98.9 *±* 0.3	98.0 *±* 0.5	98.7 *±* 0.3	98.7 *±* 0.3	98.3 *±* 0.5	98.2 *±* 0.4	99.2 *±* 0.3	98.2 *±* 0.8	97.9 *±* 0.6
	Incongruent	Response times (ms)	587 *±* 13	563 *±* 15	556 *±* 15	592 *±* 20	593 *±* 21	586 *±* 20	587 *±* 16	581 *±* 20	582 *±* 17
		Accuracy (%)	95.8 *±* 0.6	95.4 *±* 0.7	94.0 *±* 0.9	95.6 *±* 0.6	94.5 *±* 0.8	95.3 *±* 0.7	95.9 *±* 0.6	93.9 *±* 0.7	94.7 *±* 0.9

## Data Availability

Data are available on request.

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
