# Peer review of "Effect of Differing Durations of High-Intensity Intermittent Activity on Cognitive Function in Adolescents"

_ijerph, 2021, doi:10.3390/ijerph182111594_

Round 1

Reviewer 1 Report

Overall this is a well-written and interesting study that compared differing durations of high intensity intermittent aerobic activity (30 and 60 min) and rest for effects on cognitive performance among adolscents. I have only a few suggestions for improvements.

Throughout the paper, please be specific that you were testing aerobic exercise - not mixed modality (aerobic and muscle strengthening), nor muscle strengthening, but aerobic. Although the overwhelming majority of exercise research is on aerobic exercise, it is often not specified throughout articles. This small addition helps for clarity of interpretation by readers.

Abstract:

For the study purpose only the difference in exercise duration is mentioned, yet the use of high intensity intermittent aerobic exercise seemed to be a key part that also warrants mention there.

Introduction:

I did not see a hypothesis provided for the study, which I would assume the authors had a priori.

Discussion:

In the first paragraph and in your Conclusions paragraph, please specify that you tested aerobic exercise and it would be beneficial to also specific that it was running/sprinting.

What role did participant motivation play in your study findings? Running/sprinting is not motivating for many adolescents. Did you collect any information or receive any feedback about the exercise modaility used?Did all participants complete the entire 30 and 60 minute protocols?

Author Response

Overall this is a well-written and interesting study that compared differing durations of high intensity intermittent aerobic activity (30 and 60 min) and rest for effects on cognitive performance among adolescents. I have only a few suggestions for improvements.

Throughout the paper, please be specific that you were testing aerobic exercise - not mixed modality (aerobic and muscle strengthening), nor muscle strengthening, but aerobic. Although the overwhelming majority of exercise research is on aerobic exercise, it is often not specified throughout articles. This small addition helps for clarity of interpretation by readers.

Thank you for this comment. We feel that including the word aerobic when referencing the exercise undertaken may be misleading as the exercise involved sprinting, which is not strictly aerobic. We agree, however, that being more specific throughout the paper about the type of exercise performed will enhance interpretability of the findings and have therefore clarified that the high-intensity intermittent exercise involved running [Line 15, 26, 116, 122, 154, 201, 561, 565, 568, 570].

Abstract:

For the study purpose only the difference in exercise duration is mentioned, yet the use of high intensity intermittent aerobic exercise seemed to be a key part that also warrants mention there.

We agree, the use of high-intensity intermittent running exercise is an important component of the paper and have now reflected this in the study purpose [Line 14-15], thank you.

Introduction:

I did not see a hypothesis provided for the study, which I would assume the authors had a priori.

Thank you for bring this oversight to our attention. Our hypothesis for the study is now included [Line 119-122].

Discussion:

In the first paragraph and in your Conclusions paragraph, please specify that you tested aerobic exercise and it would be beneficial to also specific that it was running/sprinting.

Please see response to comment 1, thank you. We are now more specific about the form of exercise throughout the manuscript.

What role did participant motivation play in your study findings? Running/sprinting is not motivating for many adolescents. Did you collect any information or receive any feedback about the exercise modality used? Did all participants complete the entire 30 and 60 minute protocols?

We agree, not all young people are motivated by running/sprinting. Research does show, however, that young people enjoy participating in intermittent vs. continuous exercise (Malik et al., 2017) even when at high intensities (Malik et al., 2019), suggesting that the exercise modality used in the present study may be more enjoyable and motivating than continuous running/sprinting exercise. Additionally, we wanted to directly compare the effect of different durations of high-intensity intermittent running under a controlled setting, which is why we used the Loughborough Intermittent Shuttle Test because it allows for other variables (e.g. exercise intensity, physical environment, cognitive engagement) to be controlled. The findings of the study, however, can provide initial evidence for the potential effects of high-intensity intermittent running on cognition in other contexts i.e. such as in games/sports settings, which we recognise is more widely enjoyable to youth. Nevertheless, young people’s motivation for the exercise and feedback regarding the exercise was not measured as it was beyond the scope of the study and thus certainly warrants further investigation. 

All participants were able to complete the 30 min high intensity intermittent running protocol, whilst two participants were unable to complete the 60-minute protocol due to lack of fitness. The data of these participants was removed from the analysis [see methods section Line 127–128].

Reviewer 2 Report

The authors of the manuscript “Effect of differing durations of high-intensity intermittent activity on cognitive function in adolescents” presented the results of a study with 38 adolescents completed three trials separated by 7-d: 30 min exercise, 60-min exercise and Rest; in a randomized crossover design. They found that acute exercise enhanced subsequent cognition in adolescents, but overall 30-min of high-intensity intermittent exercise is more favourable to adolescents’ cognition, compared to 60-min.

The manuscript is well written, but from my perspective very long. I think it could be shortened. There are in addition some points for revision or for a critical review:

First, the major goal of this study is the effect of exercise. This is a topic for journal with a scope in sport medicine. IJERPH has other aims & scope (“the publication of scientific and technical information on the impacts of natural phenomena and anthropogenic factors on the quality of our environment, the interrelationships between environmental health and the quality of life, as well as the socio-cultural, political, economic, and legal considerations related to environmental stewardship, environmental medicine, and public health.”). So, in my opinion the manuscript doesn´t fit in the journal´s aims & scope.

The authors doesn´t described if they have checked all data for normality. The result of this test wasn´t presented. If there was no normal distribution of the data the authors should avoid to present arithmetic means and standard deviation and should presented instead of this median and interquartilrange (e.g. the presented data for age or the days spent on high intensity).

After the long introduction it is for me not really clear why the authors have used 30 and 60 minutes for their study. When I have read the introduction correctly than there is no evidence for this two times frames. This should be explained.

On line 170 the authors explained e.g. the food diary protocol. This information is not necessary for the presented data. This is one of several points were the authors can shortened their manuscript.

The authors described in several paragraph very detailed the tests that they have used. But I wasn´t able to identified how they have conducted their randomized crossover design what they mentioned in the abstract. This should be included in the method chapter.

In total I suggest to rewrite the article in regard to the points and to check if the manuscript really fit in the aims & scope of this journals.

Author Response

The authors of the manuscript “Effect of differing durations of high-intensity intermittent activity on cognitive function in adolescents” presented the results of a study with 38 adolescents completed three trials separated by 7-d: 30 min exercise, 60-min exercise and Rest; in a randomized crossover design. They found that acute exercise enhanced subsequent cognition in adolescents, but overall 30-min of high-intensity intermittent exercise is more favourable to adolescents’ cognition, compared to 60-min.

The manuscript is well written, but from my perspective very long. I think it could be shortened. There are in addition some points for revision or for a critical review:

First, the major goal of this study is the effect of exercise. This is a topic for journal with a scope in sport medicine. IJERPH has other aims & scope (“the publication of scientific and technical information on the impacts of natural phenomena and anthropogenic factors on the quality of our environment, the interrelationships between environmental health and the quality of life, as well as the socio-cultural, political, economic, and legal considerations related to environmental stewardship, environmental medicine, and public health.”). So, in my opinion the manuscript doesn´t fit in the journal´s aims & scope.

Thank you, we appreciate this and agree that it is important that a paper fits within a journal’s aims and scope. IJERPH has a key focus on public health, with three of their major sections dedicated to ‘children’s health’, ‘health promotion’ and ‘exercise and health’. Cognitive functioning impacts both physical and mental health throughout life. Moreover, IJERPH have published numerous papers on the effects of exercise on cognitive function (Farinha et al., 2021: https://doi.org/10.3390/ijerph18178963; Sok et al., 2021: https://doi.org/10.3390/ijerph18157848), including a review on the effect of acute high-intensity interval training on executive function (Ai et al., 2021: https://doi.org/10.3390/ijerph18073593). Furthermore, this paper, if accepted, will be published in IJERPH upcoming special issue on the ‘effects of physical activity on cognitive function in young people’. We therefore feel that this paper fits within the journal’s aims and scope, and especially within the scope of the targeted special issue.

The authors doesn´t described if they have checked all data for normality. The result of this test wasn´t presented. If there was no normal distribution of the data the authors should avoid to present arithmetic means and standard deviation and should presented instead of this median and interquartilrange (e.g. the presented data for age or the days spent on high intensity).

Thank you for highlighting this. The steps taken to check the normality of the data distribution has been clarified within the methods section of the paper [Line 282–285].

After the long introduction it is for me not really clear why the authors have used 30 and 60 minutes for their study. When I have read the introduction correctly than there is no evidence for this two times frames. This should be explained.

Thank you for this comment. Currently, the findings of reviews and meta-analyses regarding the optimum and minimum duration of exercise necessary for enhancements to cognition are contradictory, with some concluding that shorter durations are more effective and others suggesting that shorter durations are insufficient, with longer durations needed. Considering the findings of the reviews/meta-analyses together, a duration of between 10-30 min may be most favourable for enhancing subsequent cognition. The two studies to date which have compared exercise durations found no effects to cognition from 5, 10, 20 or 30 min, but this may have been due to the type of exercise not being of a high enough intensity to elicit effects at these shorter durations. We therefore felt it would be useful to see whether 30 min of exercise would be beneficial if it was of a high-intensity. Moreover, a previous study which utilised high-intensity exercise found that 60 min enhanced both immediate and delayed (45 min post exercise) cognition in young people. We thus wanted to compare 30 min to 60 min of high-intensity intermittent exercise. Overall, as primary research comparing the effects of different durations of exercise is lacking, we believed this would be a good place to start. Nevertheless, it would be beneficial for future research to compare different durations of high-intensity intermittent running e.g. 10 vs 20 vs 30 min. This has been highlighted within the discussion [Line 528–532] and conclusion [Line 567–568] section of the paper.

On line 170 the authors explained e.g. the food diary protocol. This information is not necessary for the presented data. This is one of several points were the authors can shortened their manuscript.

Thank you, we appreciate this suggestion. However, as diet influences cognitive function directly and influences the effects of exercise on cognitive function, it is vital that diet is controlled in studies of this nature. This information has thus been included to demonstrate that this potentially extraneous variable has not influenced the results and to enable (and encourage) researchers to replicate this method of dietary control in future, when conducting research of this nature.

The authors described in several paragraph very detailed the tests that they have used. But I wasn´t able to identified how they have conducted their randomized crossover design what they mentioned in the abstract. This should be included in the method chapter.

Thank you for highlighting this. The Latin Square Design was used to randomise the order in which participants completed the three main trials. This information has now been included in the methods section [Line 155–156].

In total I suggest to rewrite the article in regard to the points and to check if the manuscript really fit in the aims & scope of this journals.

We have shortened the paper where possible and feel that the paper has been improved as a result of your comments, thank you. We feel that the paper fits within the ‘child health’ and ‘exercise and health’ section of the journal and feel it would sit nicely within the journal’s upcoming special issue on the ‘effects of physical activity on cognitive function in young people’.

Reviewer 3 Report

TITLE

1. Is clear and well constructed

ABSTRACT
is clear and well written

MAIN TEXT

An article is well written

MINOR COMMENTS

Authors should discuss with some ‘fresh’ articles

  1. In the first paragraph of Intro I suggest to add 1-2 sentences inndicating tha Authors know some neuro paths:
  1. a) "the dorsolateral prefrontal cortex, sensory motor area and superior temporal gyrus are activated predominantly during static and dynamic balance tasks"

ABHINAV SATHE, SHWETA SHENOY, PRACHI KHANDEKAR SATHE, Evaluation of cerebral cortex activation during balance tasks
using fNIRS: a systematic review. TRENDS in
Sport Sciences
2021; 28(2): 93-107 http://tss.awf.poznan.pl/files/2021/Vol%2028%20no%202/3_Sathe_TSS_2021_282_93-107.pdf

  1. b) In Discussion I suggest reffer to Cognicise - a new model of exercise

more u find here

http://tss.awf.poznan.pl/files/2021/Vol%2028%20no%201/1_GRONEK_TSS_2021_281_5-10.pdf

Conlcusions
The article includes very valuable data with good practical outcome. The list of reference is a little not appropriate and in my opinion should contain article mentioned above.

Author Response

TITLE

1. Is clear and well constructed

 ABSTRACT
is clear and well written

MAIN TEXT

 An article is well written

MINOR COMMENTS

 Authors should discuss with some ‘fresh’ articles

  1. In the first paragraph of Intro I suggest to add 1-2 sentences inndicating tha Authors know some neuro paths:
  1. a) "the dorsolateral prefrontal cortex, sensory motor area and superior temporal gyrus are activated predominantly during static and dynamic balance tasks"

ABHINAV SATHE, SHWETA SHENOY, PRACHI KHANDEKAR SATHE, Evaluation of cerebral cortex activation during balance tasks
using fNIRS: a systematic review. TRENDS in
Sport Sciences
2021; 28(2): 93-107 http://tss.awf.poznan.pl/files/2021/Vol%2028%20no%202/3_Sathe_TSS_2021_282_93-107.pdf

Thank you for your suggestion and for bringing this interesting paper to our attention. We have now highlighted some of the mediating variables involved in the exercise-cognition relationship, and have referenced this article, in the introduction of the paper [Line 36–39].

  1. b) In Discussion I suggest reffer to Cognicise - a new model of exercise

more u find here

http://tss.awf.poznan.pl/files/2021/Vol%2028%20no%201/1_GRONEK_TSS_2021_281_5-10.pdf

Thank you for this suggestion. This certainly is an interesting new model of exercise and the effects of cognitively-engaging exercise on children’s cognition is also gaining increasing research attention. We have highlighted this as an important area for future research in the discussion section of the paper [Line 553–557].

Conclusions
The article includes very valuable data with good practical outcome. The list of reference is a little not appropriate and in my opinion should contain article mentioned above.

Thank you for this suggestion, both articles suggested above have now been included in the paper and in the reference list [Line 613–614, 728–729].

Round 2

Reviewer 2 Report

None of my comments from my first review was considered for this new version. So I have to repeat my points from this review:

The manuscript is well written, but from my perspective very long. I think it could be shortened. There are in addition some points for revision or for a critical review:

First, the major goal of this study is the effect of exercise. This is a topic for journal with a scope in sport medicine. IJERPH has other aims & scope (“the publication of scientific and technical information on the impacts of natural phenomena and anthropogenic factors on the quality of our environment, the interrelationships between environmental health and the quality of life, as well as the socio-cultural, political, economic, and legal considerations related to environmental stewardship, environmental medicine, and public health.”). So, in my opinion the manuscript doesn´t fit in the journal´s aims & scope.

The authors doesn´t described if they have checked all data for normality. The result of this test wasn´t presented. If there was no normal distribution of the data the authors should avoid to present arithmetic means and standard deviation and should presented instead of this median and interquartilrange (e.g. the presented data for age or the days spent on high intensity).

After the long introduction it is for me not really clear why the authors have used 30 and 60 minutes for their study. When I have read the introduction correctly than there is no evidence for this two times frames. This should be explained.

On line 170 the authors explained e.g. the food diary protocol. This information is not necessary for the presented data. This is one of several points were the authors can shortened their manuscript.

The authors described in several paragraph very detailed the tests that they have used. But I wasn´t able to identified how they have conducted their randomized crossover design what they mentioned in the abstract. This should be included in the method chapter.

In total I suggest to rewrite the article in regard to the points and to check if the manuscript really fit in the aims & scope of this journals.

Author Response

Apologies, I am not sure if there was a technical issue, but we did respond to your comments as per the below. I have copied and pasted them for you here - hopefully this works this time!!

The authors of the manuscript “Effect of differing durations of high-intensity intermittent activity on cognitive function in adolescents” presented the results of a study with 38 adolescents completed three trials separated by 7-d: 30 min exercise, 60-min exercise and Rest; in a randomized crossover design. They found that acute exercise enhanced subsequent cognition in adolescents, but overall 30-min of high-intensity intermittent exercise is more favourable to adolescents’ cognition, compared to 60-min.

The manuscript is well written, but from my perspective very long. I think it could be shortened. There are in addition some points for revision or for a critical review:

First, the major goal of this study is the effect of exercise. This is a topic for journal with a scope in sport medicine. IJERPH has other aims & scope (“the publication of scientific and technical information on the impacts of natural phenomena and anthropogenic factors on the quality of our environment, the interrelationships between environmental health and the quality of life, as well as the socio-cultural, political, economic, and legal considerations related to environmental stewardship, environmental medicine, and public health.”). So, in my opinion the manuscript doesn´t fit in the journal´s aims & scope.

Thank you, we appreciate this and agree that it is important that a paper fits within a journal’s aims and scope. IJERPH has a key focus on public health, with three of their major sections dedicated to ‘children’s health’, ‘health promotion’ and ‘exercise and health’. Cognitive functioning impacts both physical and mental health throughout life. Moreover, IJERPH have published numerous papers on the effects of exercise on cognitive function (Farinha et al., 2021: https://doi.org/10.3390/ijerph18178963; Sok et al., 2021: https://doi.org/10.3390/ijerph18157848), including a review on the effect of acute high-intensity interval training on executive function (Ai et al., 2021: https://doi.org/10.3390/ijerph18073593). Furthermore, this paper, if accepted, will be published in IJERPH upcoming special issue on the ‘effects of physical activity on cognitive function in young people’. We therefore feel that this paper fits within the journal’s aims and scope, and especially within the scope of the targeted special issue.

The authors doesn´t described if they have checked all data for normality. The result of this test wasn´t presented. If there was no normal distribution of the data the authors should avoid to present arithmetic means and standard deviation and should presented instead of this median and interquartilrange (e.g. the presented data for age or the days spent on high intensity).

Thank you for highlighting this. The steps taken to check the normality of the data distribution has been clarified within the methods section of the paper [Line 282–285].

After the long introduction it is for me not really clear why the authors have used 30 and 60 minutes for their study. When I have read the introduction correctly than there is no evidence for this two times frames. This should be explained.

Thank you for this comment. Currently, the findings of reviews and meta-analyses regarding the optimum and minimum duration of exercise necessary for enhancements to cognition are contradictory, with some concluding that shorter durations are more effective and others suggesting that shorter durations are insufficient, with longer durations needed. Considering the findings of the reviews/meta-analyses together, a duration of between 10-30 min may be most favourable for enhancing subsequent cognition. The two studies to date which have compared exercise durations found no effects to cognition from 5, 10, 20 or 30 min, but this may have been due to the type of exercise not being of a high enough intensity to elicit effects at these shorter durations. We therefore felt it would be useful to see whether 30 min of exercise would be beneficial if it was of a high-intensity. Moreover, a previous study which utilised high-intensity exercise found that 60 min enhanced both immediate and delayed (45 min post exercise) cognition in young people. We thus wanted to compare 30 min to 60 min of high-intensity intermittent exercise. Overall, as primary research comparing the effects of different durations of exercise is lacking, we believed this would be a good place to start. Nevertheless, it would be beneficial for future research to compare different durations of high-intensity intermittent running e.g. 10 vs 20 vs 30 min. This has been highlighted within the discussion [Line 528–532] and conclusion [Line 567–568] section of the paper.

On line 170 the authors explained e.g. the food diary protocol. This information is not necessary for the presented data. This is one of several points were the authors can shortened their manuscript.

Thank you, we appreciate this suggestion. However, as diet influences cognitive function directly and influences the effects of exercise on cognitive function, it is vital that diet is controlled in studies of this nature. This information has thus been included to demonstrate that this potentially extraneous variable has not influenced the results and to enable (and encourage) researchers to replicate this method of dietary control in future, when conducting research of this nature.

The authors described in several paragraph very detailed the tests that they have used. But I wasn´t able to identified how they have conducted their randomized crossover design what they mentioned in the abstract. This should be included in the method chapter.

Thank you for highlighting this. The Latin Square Design was used to randomise the order in which participants completed the three main trials. This information has now been included in the methods section [Line 155–156].

In total I suggest to rewrite the article in regard to the points and to check if the manuscript really fit in the aims & scope of this journals.

We have shortened the paper where possible and feel that the paper has been improved as a result of your comments, thank you. We feel that the paper fits within the ‘child health’ and ‘exercise and health’ section of the journal and feel it would sit nicely within the journal’s upcoming special issue on the ‘effects of physical activity on cognitive function in young people’.

Round 3

Reviewer 2 Report

The manuscript is well written, even the authors have responsed to my first review comments, they didn´t have take into change the manuscript as they have to do in my opinion to make the article acceptable.

From my perspective the article is still to long. I have recommened to shorten the article and see there is many options to do this. Instead of this the article have become longer from the first version (18 pages) to the last one (19 pages).

There is still a important mistake in the statistical analysis. The authors doesn´t described if they have checked all data for normality. The result of this test wasn´t presented. If there was no normal distribution of the data the authors should avoid to present arithmetic means and standard deviation and should presented instead of this median and interquartilrange (e.g. the presented data for age or the days spent on high intensity). For example, when I look on the decribed data about the age of the persons examined then the SD is so small that I am very sure, that this data doesn´t have a normal distribution. This have to be checked for ALL decribed data.

The authors described in several paragraph very detailed the tests that they have used. This is a example where the manuscript can be shorten very easy. In the second version they have no included their randomized crossover design, but I wonder why the authors sometimes referred to several studies to underline there study design and in this important point they didn´t use any reference. This is one example how the manuscript can be improved significantly.

In total I suggest to rewrite the article in regard to the points. The article have to be shorten significantly and the reference should be proofed if they are really important for this article.

Author Response

Please see Word file attached
